# Efficient Inhibition of Deep Conversion of Partial Oxidation Products in C-H Bonds’ Functionalization Utilizing O_2_ via Relay Catalysis of Dual Metalloporphyrins on Surface of Hybrid Silica Possessing Capacity for Product Exclusion

**DOI:** 10.3390/biomimetics9050272

**Published:** 2024-04-29

**Authors:** Yu Zhang, Xiao-Ling Feng, Jia-Ye Ni, Bo Fu, Hai-Min Shen, Yuan-Bin She

**Affiliations:** 1College of Chemical Engineering, Zhejiang University of Technology, Hangzhou 310014, China; lym15665861061@163.com (Y.Z.); njy970615@163.com (J.-Y.N.); fubo13479347998@163.com (B.F.); 2Hangzhou Copiore Chemical Technology Co., Ltd., Hangzhou 310012, China; sales@copiore.com

**Keywords:** C-H bonds, hydrocarbons, oxidation, partial oxidation, oxygen, relay catalysis

## Abstract

To inhibit the deep conversion of partial oxidation products (POX-products) in C-H bonds’ functionalization utilizing O_2_, 5-(4-(chloromethyl)phenyl)-10,15,20-tris(perfluorophenyl)porphyrin cobalt(II) and 5-(4-(chloromethyl)phenyl)-10,15,20-tris(perfluorophenyl)porphyrin copper(II) were immobilized on the surface of hybrid silica to conduct relay catalysis on the surface. Fluorocarbons with low polarity and heterogeneous catalysis were devised to decrease the convenient accessibility of polar POX-products to catalytic centers on the lower polar surface. Relay catalysis between Co and Cu was designed to utilize the oxidation intermediates alkyl hydroperoxides to transform more C-H bonds. Systematic characterizations were conducted to investigate the structure of catalytic materials and confirm their successful syntheses. Applied to C-H bond oxidation, not only deep conversion of POX-products was inhibited but also substrate conversion and POX-product selectivity were improved simultaneously. For cyclohexane oxidation, conversion was improved from 3.87% to 5.27% with selectivity from 84.8% to 92.3%, which was mainly attributed to the relay catalysis on the surface excluding products. The effects of the catalytic materials, product exclusion, relay catalysis, kinetic study, substrate scope, and reaction mechanism were also investigated. To our knowledge, a practical and novel strategy was presented to inhibit the deep conversion of POX-products and to achieve efficient and accurate oxidative functionalization of hydrocarbons. Also, a valuable protocol was provided to avoid over-reaction in other chemical transformations requiring high selectivity.

## 1. Introduction

The functionalization of C-H bonds by means of oxidation utilizing molecular oxygen (O_2_) is an extensively adopted strategy in the industrial utilization of fossil resources containing abundant hydrocarbons [1,2,3,4,5]. And, a troublesome issue in these important industrial processes is the deep conversion of partial oxidation products (POX-products) because POX-products, such as alcohols and ketones, are usually more reactive than substrates like cycloalkanes, alkanes, and alkyl aromatics [6,7,8,9]. Thus, in the field of C-H bonds’ oxidation utilizing O_2_, a widely acknowledged fact is that partial oxidation products will be converted further to deep oxidation products along with their formations, especially at high conversion levels, resulting in consumed selectivity towards partial oxidation products induced by improved substrate conversion. In order to obtain POX-products with a satisfying selectivity, a widely applied strategy is to lower the substrate conversion. For example, in the industrial oxidation of cyclohexane with O_2_ to produce the POX-product KA oil, comprising cyclohexanol and cyclohexanone, which not only are the fundamental intermediates in the industrial manufacturing of polyamide fiber Nylon-66 and engineering plastics but also the essential raw materials in syntheses of other valuable fine chemicals, the substrate conversion is usually controlled in the range of 5–8% with the purpose of achieving an acceptable selectivity of 75–85% towards the POX-product KA oil [10,11,12,13]. The formed by-products mainly contain hexanedioic acid (HOOC(CH_2_)_4_COOH), pentanedioic acid (HOOC(CH_2_)_3_COOH), and their cyclohexyl esters, which not only complicate the separation processes and increase the separation energy consumption in industrial hydrocarbon oxidation but also increase the waste discharge [9,14,15]. In addition, these by-products could also induce plugging of the pipeline in the continuous flow reactor. Moreover, in such low substrate conversion, mass unconverted hydrocarbons need to be recovered and recycled. It is very obvious that the propensity of partial oxidation products to be deep-oxidized makes C-H bonds’ oxidation utilizing O_2_ a process with high energy consumption and high pollution in the current chemical industry. Considering the requirements for green chemistry [16,17,18,19], efficient inhibition of the deep conversion of partial oxidation products becomes an urgent demand in C-H bonds’ functionalization utilizing O_2_ not only in academic research but also in industrial applications. It is best to realize the inhibition of deep oxidation of partial oxidation products efficiently with increased conversion.

To inhibit the deep conversion of POX-products in C-H bonds’ oxidation utilizing O_2_ and realize efficient and accurate hydrocarbon oxidation, several intelligent and effective protocols have been explored and reported [20,21,22,23,24,25]. Based on the difference in research focus, these protocols can be mainly divided into two categories. In the first category, novel catalytic materials are paid more attention, and they are the research focus, in which the efficient inhibition of the deep conversion of POX-products is regarded as relying on efficient catalytic materials and the conversion of substrates to POX-products is treated as a whole process. For example, employing Ce-Fe oxides derived from Prussian blue analogues as catalysts, Liu and collaborators conducted selective oxidation of cyclohexane with O_2_ [21]. At 150 °C and under 1.0 MPa O_2_, a higher conversion of 9.03% was obtained compared to the current industrial level in the selectivity of 81.8% towards the POX-product KA oil (cyclohexanol and cyclohexanone). When the pressure of O_2_ was raised from 1.0 MPa to 1.5 MPa, the conversion rose to 11.9%, but the selectivity towards POX-products dropped down to 78.4%. Meanwhile, the selectivity of deep oxidation products and others climbed from 18.2% to 21.6%. An obvious deep conversion of partial oxidation products along with increased conversion was observed. Recently, Yuan and co-workers reported atomically dispersed cobalt on graphitic carbon nitride as an effective catalyst applied in the efficient oxidative conversion of cyclohexane with O_2_ [6]. After reacting for 6.0 h at 130 °C under a pressure of 0.80 MPa, a high substrate conversion of 9.1% was achieved with a selectivity of 59% towards KA oil, and the selectivity towards deep oxidation product 1,6-hexanedioic acid reached up to 38%. And, as the substrate was shifted to toluene, a selectivity of 83% towards deep-oxidation product benzoic acid was achieved in a conversion of 77%. The conversion and selectivity reported in their work are a set of relatively realistic and reasonable experimental data regarding hydrocarbons’ oxidation. In high substrate conversion, KA oil would inevitably be deep-oxidized because of its high reactivity, and it is a very difficult challenge to inhibit the deep conversion of POX-products in C-H bonds’ oxidation, especially when substrate conversion is at a higher level. Besides traditional thermal catalysis, Wang and cooperators applied photothermal catalysis to the oxidation of cyclohexane [20]. In their work, Co_8_La_1_O_x_/WO_3_ Z-scheme nanocomposites were developed as catalysts, and a high conversion of 10.4% was obtained with selectivity of 81.0% towards KA oil and 17.8% towards deep oxidation product 1,6-hexanedioic acid in the conditions of 1.50 MPa O_2_, 120 °C, and irradiation with a 400 nm light. Based on this search of the literature, it is very obvious that the utilization of novel and effective catalysts enhanced the conversion efficiency of C-H bonds’ oxidation utilizing O_2_, but the deep conversion of partial oxidation products still cannot be inhibited effectively for their higher reactivity.

In the second category of strategies to inhibit deep oxidation and realize efficient and accurate C-H bond functionalization utilizing O_2_, the oxidation of hydrocarbons has been explored by proceeding through three stages, which are demonstrated in Figure 1, rather than a whole process [9,14,15,26,27,28,29]. In the first stage (Black Line), C-H bonds are converted to important oxidation intermediates, alkyl hydroperoxides, through autoxidation, catalytic oxidation, or their combination depending on the reaction temperature and the presence of catalysts. Then, in the second stage (Blue Line), alkyl hydroperoxides are transformed to partial oxidation products, alcohols and ketones, through thermal decomposition, catalytic conversion, or their combination. In the thermal decomposition of hydroperoxides, a great quantity of reactive free radicals are produced, resulting in the generated partial oxidation products being attacked frequently by reactive free radicals and being deep-oxidized. On the other hand, in the catalytic conversion route, hydroperoxides are smoothly transformed to POX-products, and the deep oxidation of formed POX-products can be inhibited effectively. And, the third stage (Pink Line) is the deep oxidation of partial oxidation products through autoxidation or catalytic transformation. In current research on partial oxidation of hydrocarbons utilizing O_2_, the first stage and second stage are usually taken as one whole, and not enough attention has been paid to intensifying the catalytic transformation of oxidation intermediates alkyl hydroperoxides. In our preliminary exploration to inhibit the deep conversion in C-H bonds’ functionalization utilizing O_2_ and realize efficient and accurate hydrocarbon functionalization utilizing O_2_, practical and original relay catalysis modes were developed based on two designed metal centers with a focus on the catalytic conversion of alkyl hydroperoxides and utilizing their intrinsic oxidability [9,14,15,26,27,28]. Because hydroperoxides are transformed in the presence of catalysts, not only the deep oxidation of POX-products was inhibited but also the substrate conversion was heightened, resulting in enhanced selectivity towards POX-products along with increased substrate conversion. For example, with 2D metalloporphyrinic MOFs Co-TCPPCu as the relay catalyst and cyclohexane as the model substrate, the selectivity towards POX-products climbed up to 95% from 85%, with substrate conversion climbing up to 5.31% from 4.39% compared to homogeneous metalloporphyrin T(4-COOCH_3_)PPCo with a single metal center as the catalyst, illustrating the effective inhibition of the deep oxidation of partial oxidation products [15]. In the above relay catalysis, the metal center cobalt was utilized to activate O_2_, and the second metal center, comprising copper or others, was utilized to boost the catalytic conversion of the oxidation intermediates hydroperoxides.

Relay catalysis conversion has become a very effective and rational strategy to inhibit deep oxidation and realize efficient and accurate C-H bond functionalization utilizing O_2_ [9,14,15,27,28]. To further proceed towards the efficient and accurate oxidation of C-H bonds utilizing O_2_, in our recent research, it was found that besides the transformation modes of hydroperoxides, the convenient accessibility of POX-products to catalytic active centers was another source of deep oxidation. To accelerate the disengagement of POX-products from catalytic active centers and suppress their accessibility to catalytic active centers, it may be another effective strategy to inhibit deep oxidation in the oxidative functionalization of C-H bonds utilizing O_2_. Thus, 5-(4-(chloromethyl)phenyl)-10,15,20-tris(perfluorophenyl)porphyrin cobalt(II) (abbreviated as Tris(perF)P(4-CH_2_Cl)PPCo or Porp. Co) were immobilized on hybrid silica with Tris(perF)P(4-CH_2_Cl)PPCu (Porp. Cu), Tris(perF)P(4-CH_2_Cl)PPZn (Porp. Zn), and Tris(perF)P(4-CH_2_Cl)PPNi (Porp. Ni), respectively, to construct a bimetallic relay catalysis model (Si@Porp. Co&Cu, Si@Porp. Co&Zn, Si@Porp. Co&Ni) in this work, as illustrated in Figure 2. And, the heterogeneous catalysis was employed as the first measure to decrease the convenient accessibility of POX-products to catalytic active centers. More importantly, the C-F bonds in metalloporphyrins were devised as the second measure to accelerate the disengagement of partial oxidation products from catalytic active centers and suppress their accessibility, both of which originate from the low affinity and wettability between polar partial oxidation products (alcohols and ketones) and less-polar fluorocarbons [30,31,32]. Applied to C-H bond oxidation utilizing O_2_ as an oxygen source and cyclohexane as a model substrate, the deep conversion of partial oxidation products was inhibited effectively indeed (Figure 2). Compared with homogeneous 5,10,15,20-tetrakis(pentafluorophenyl)porphyrin cobalt(II) (T(perF)PPCo) as a catalyst, the selectivity climbed to 92.3% from 85.2% for POX-products; furthermore, the substrate conversion rose to 5.27% from 4.25% in mild conditions (120 °C, 1.0 MPa O_2_). The deep conversion of partial oxidation products was indeed inhibited through the relay catalysis of dual metalloporphyrins on the surface of hybrid silica possessing the capacity for product exclusion. The relay catalytic mechanism, product exclusion performance, apparent kinetics, substrate scope, and reaction mechanism in C-H bonds’ oxidation utilizing O_2_ were also explored systematically. Compared with the current literature, the protocol developed in this work is a very novel, intelligent, and applicable strategy and mode to realize efficient and accurate C-H bond oxidation utilizing O_2_ through relay catalysis of bimetallic centers on the surface of hybrid silica with the capacity for product exclusion. And, the process of C-H bonds’ oxidation from substrate to partial oxidation products was considered and adjusted comprehensively as two sequential and separate processes in this work, which presented a novel and practical relay catalysis model on the surface of catalytic materials. Therefore, in this work, not only a useful, applicable, and wise mode and reference are provided for selective C-H bonds’ functionalization utilizing O_2_ in academic research and industrial applications but also a referential research case for the design of other catalysis systems requiring high selectivity and high efficiency.

## 2. Experimental Section

The information about chemical reagents used in experiments, the characterization details and corresponding instruments, the syntheses of porphyrins and metalloporphyrins, the preparation of hybrid silica (Si@NH_2_), the apparent kinetics study, the free radical capture experiments, and the EPR measurement are presented in the Appendix A [9,15,26,33]. The ^1^H NMR, ^13^C NMR and ESI-MS spectra were presented in the Appendix A as Appendix A.

### 2.1. Immobilization of Metalloporphyrins on Hybrid Silica

The immobilization of metalloporphyrins on hybrid silica (Si@NH_2_) was accomplished through the nucleophilic substitution between –NH_2_ in hybrid silica and –CH_2_Cl in metalloporphyrins. Taking the optimized catalytic material as an example (Si@Porp. Co&Cu), the synthesis procedure was conducted as follows. In a glass reaction tube (50 mL), hybrid silica (Si@NH_2_) (0.5000 g), 5-(4-(chloromethyl)phenyl)-10,15,20-tris(perfluorophenyl)porphyrin cobalt(II) (Tris(perF)P(4-CH_2_Cl)PPCo) (0.0264 g, 0.0267 mmol), 5-(4-(chloromethyl)phenyl)-10,15,20-tris(perfluorophenyl)porphyrin copper(II) (Tris(perF)P(4 -CH_2_Cl)PPCu) (0.0530 g, 0.0533 mmol), anhydrous potassium carbonate (K_2_CO_3_, 0.0111 g, 0.08 mmol), and potassium iodide (KI, 0.0017 g, 0.01 mmol) were dissolved in dry DMF (20 mL) with stirring and the protection of nitrogen. The temperature of the reaction mixture was raised to 80 °C and kept constant with stirring for 48.0 h under N_2_. Then, a pink solid collected through centrifugation from the reaction mixture at room temperature was washed successively with dry DMF (5 × 8 mL), distilled water (5 × 8 mL), and acetone (5 × 6 mL) until the upper liquid became clear and colorless. After being dried at 60 °C under reduced pressure for 12.0 h, the immobilized dual metalloporphyrin was obtained as a pink power (0.1346 g). The same procedures were utilized to prepare other immobilized dual metalloporphyrins (Si@Porp. Co&Zn, Si@Porp. Co&Ni, Si@Porp. Cu&Zn) and immobilized mono metalloporphyrins (Si@Porp. Co, Si@Porp. Cu, Si@Porp. Zn).

### 2.2. Partial Oxidation of C-H Bonds with O_2_

In this work, substrates possessing C-H bonds were selected as cycloalkanes and alkyl aromatics. The partial oxidation utilizing O_2_ was conducted in a high-pressure reactor made from stainless steel equipped with an inner lining made from polytetrafluoroethylene. Immobilized metalloporphyrin (0.0320 g) as a catalyst and substrate (200 mmol) were mixed in a high-pressure reactor (100 mL) and the reactor was sealed. Then, the temperature of the reaction mixture was raised to the target value in an oil bath with stirring, such as 120 °C. Molecular oxygen (O_2_) flowed into the sealed high-pressure reactor to raise the reaction pressure to the target value. The reaction was conducted for a set time with stirring at the target temperature and under the target pressure. The reaction mixture was transferred to an ice-water bath to be cooled to room temperature quickly as the reaction time was reached. At room temperature, the high-pressure reactor was opened, and triphenylphosphine (1.3115 g, 5.00 mmol) was added quickly with additional stirring for 30.0 min with the purpose of reducing the formed hydroperoxides to alcohols. At last, the reaction mixture was dissolved in acetone (100 mL) exactly. For quantitative analyses of the oxidation products, 10 mL of the obtained acetone solution was utilized to carry out GC analyses with toluene or naphthalene as the internal standard (2.0 mmol). Another 10 mL was used in HPLC analyses with benzoic acid or 2-naphthoic acid as the internal standard (0.20 mmol). All of the catalytic experiments were conducted more than twice, and the obtained experimental results were the averages of these experiments.

### 2.3. Relay Catalysis Study

To explore the relay catalysis mechanism, the C-H bonds’ oxidation by alkyl hydroperoxides (R-OOH) was validated utilizing cyclopentane as a model hydrocarbon containing C-H bonds and cyclohexyl hydroperoxides as the oxidant. In the relay catalysis study, metalloporphyrins Tris(perF)P(4-CH_2_Cl)PPCo and Tris(perF)P(4-CH_2_Cl)PPCu and their physical mixture optimized the immobilized metalloporphyrins Si@Porp. Co, Si@Porp. Cu, and Si@Porp. Co&Cu were utilized as catalysts. The experimental procedure was conducted as follow. Cyclohexyl hydroperoxides in cyclohexane obtained from the autoxidation of cyclohexane utilizing O_2_ at 135 °C under the pressure of 1.0 MPa for 8.0 h (about 1.5%, mol/mol, 5.0 mL, 3.9770 g) were mixed with triphenylphosphine (0.5246 g, 2.00 mmol) and stirred for 30.0 min to analyze the concentration of cyclohexyl hydroperoxides and other partial oxidation products, as illustrated in 2.2. Then, in a new and clean glass reaction tube (25 mL), cyclohexyl hydroperoxides in cyclohexane obtained from the autoxidation of cyclohexane at 135 °C and 1.0 MPa O_2_ for 8.0 h (about 1.5%, mol/mol, 5.0 mL, 3.9770 g), cyclopentane (0.7014 g, 10 mmol), and a catalyst (metalloporphyrins, 0.0020 g, immobilized metalloporphyrins, 0.0320 g) or no catalyst were mixed through stirring. The temperature of the reaction mixture was raised to 60 °C with stirring under the protection of nitrogen. After reacting for 6.0 h, the reaction mixture was transferred to an ice-water bath to be cooled to room temperature. At room temperature, triphenylphosphine (0.1311 g, 0.50 mmol) was added quickly to the reaction mixture with additional stirring for 30.0 min with the purpose of reducing the residual hydroperoxides. The reaction mixture was dissolved in acetone exactly (10 mL) to determine the conversions of cyclohexyl hydroperoxides and cyclopentane and the selectivity of cyclopentanol and cyclopentanone (Partial oxidation products of cyclopentane) by means of GC and HPLC, as illustrated in Section 2.2.

### 2.4. UV-Vis Measurement

Metalloporphyrins, such as 5-(4-(chloromethyl)phenyl)-10,15,20-tris(perfluorophenyl)porphyrin cobalt(II) (Tris(perF)P(4-CH_2_Cl)PPCo), Tris(perF)P(4-CH_2_Cl)PPCu, or Tris(perF)P(4-CH_2_Cl)PPZn (0.002 mmol), were dissolved in dry and redistilled cyclohexane (100 mL), promoted by ultrasound at room temperature and under the atmospheric pressure. The time of ultrasound promotion was exactly 5.0 min. Under the atmospheric pressure, two milliliters of the obtained solution was quickly transferred to a quartz cuvette to carry out UV-Vis measurements at room temperature. The UV-Vis spectra were collected every 5.0 min in the adsorption range of 200–800 nm to explore the change in UV-Vis absorption of the solution.

### 2.5. Contact Angle Measurement

The contact angles of substrates and their oxidation products on the surface of immobilized metalloporphyrins were measured utilizing cyclohexane and its oxidation products cyclohexanol and cyclohexanone as model compounds. The immobilized metalloporphyrins, such as optimized Si@Porp. Co&Cu or hybrid silica (Si@NH_2_) (0.0500 g), were compressed into flakes in a tableting device under the pressure of 2.50 MPa for 1.0 min. Cyclohexane, cyclohexanol, or cyclohexanone (2.0 µL) was dripped onto the surface of the obtained catalytic material tableting. After the droplet stabilized on the surface for 20 s, the contact angle of the droplet on the catalytic material was measured and obtained using the KRÜSS DSA 100S Automatic Contact Angle Measuring Instrument.

## 3. Results and Discussion

### 3.1. Characterization

The immobilization of metalloporphyrins on the surface of hybrid silica (Si@NH_2_) was firstly confirmed through FT-IR (Fourier transform infrared spectroscopy) measurements. It was demonstrated in Figure 1 that when metalloporphyrins, such as Tris(perF)P(4-CH_2_Cl)PPCo and Tris(perF)P(4-CH_2_Cl)PPCu, were immobilized on Si@NH_2_, characteristic absorption peaks of metalloporphyrins around 2930 cm^−1^ (light blue oblique line area), 1650 cm^−1^ (black dots), and 1520 cm^−1^ (red dots), which were attributed to the stretching vibration of C-H bonds, C=N bonds in the porphyrin ring, and C=C bonds in the benzene ring, respectively, emerged in the FT-IR spectra of Si@Porp. Co&Cu, Si@Porp. Co&Zn, and Si@Porp. Co compared with hybrid silica (Si@NH_2_). And, the characteristic absorption peaks of Si@NH_2_ at about 3250 cm^−1^ (green oblique line area), 1050 cm^−1^ (blue dots), and 780 cm^−1^ (pink dots), which belonged to the stretching vibration of O-H and N-H bonds, Si-O bonds, and Si-C bonds, respectively, were reserved in the immobilized metalloporphyrins (Si@Porp. Co&Cu, Si@Porp. Co&Zn, and Si@Porp. Co). Compared with isolated metalloporphyrins, all of the characteristic absorption peaks of metalloporphyrins were retained in the immobilized metallloporphyrins, and the immobilization did not affect the structure of metalloporphyrins. It could be observed clearly from the FT-IR measurements that metalloporphyrins were immobilized on the surface of hybrid silica (Si@NH_2_) successfully, and the main structure of Si@NH_2_ was not affected significantly by the immobilization of metalloporphyrins. The successful immobilization of metalloporphyrins on the surface of Si@NH_2_ was also proven through the XPS (X-ray photoelectron spectroscopy) analyses. In the total XPS spectra of Si@Porp. Co&Cu, Si@Porp. Co&Zn, and Si@Porp. Co (Figure 2), not only the characteristic elements of Si@NH_2_ (Si, C, N, and O) were detected but also the representative elements of metalloporphyrins (N, Co, Cu, Zn, and F) were obviously observed, especially in the enlarged high-resolution XPS spectra. It was also found from the high-resolution XPS analyses obtained through the software XPS Peak41 that the cobalt element in Si@Porp. Co&Cu, Si@Porp. Co&Zn, and Si@Porp. Co, which was located in 797 eV and 782 eV as Co 2p_1/2_ and Co 2p_3/2_, respectively, mainly comprised two parts, +2 valence and +3 valence [34,35,36]. And, the dominant valence state of cobalt in the immobilized metalloporphyrins was +2 valence in this work. As for the copper and zinc elements in Si@Porp. Co&Cu and Si@Porp. Co&Zn, which were located in 956 eV and 936 eV as Cu 2p_1/2_ and Cu 2p_3/2_, respectively [37,38,39], and 1046 eV and 1023 eV as Zn 2p_1/2_ and Zn 2p_3/2_, respectively [40,41,42], they exhibited only one valence state as +2. The existence of +3 valence in cobalt elements was not only consistent with its characteristic as a variable valence metal but also implied the ability of cobalt in +2 valence to bind with and activate molecular oxygen thermodynamically and spontaneously, which was the source of the catalytic performance exhibited by the immobilized metalloporphyrins containing cobalt in +2 valence.

The crystallographic structures of the immobilized metalloporphyrins and metalloporphyrin unit (Tris(perF)P(4-CH_2_Cl)PPCo) in this work were explored using the PXRD (Powder X-ray diffraction) technique. As demonstrated in Figure 3, both the hybrid silica (Si@NH_2_) and immobilized metalloporphyrins (Si@Porp. Co&Cu, Si@Porp. Co&Zn, and Si@Porp. Co) exhibited a broad diffraction peak at about 23°, and no obvious crystal diffraction peak was observed, implying that all of Si@NH_2_, Si@Porp. Co&Cu, Si@Porp. Co&Zn, and Si@Porp. Co were amorphous powder. As for the extra and very small peaks around 18, 35, and 45 degrees, they might be the impurity peaks, and they would not affect the catalytic performance of immobilized metalloporphyrins. The PXRD measurements also provided strong evidence that the immobilization of metalloporphyrins on the surface of hybrid silica (Si@NH_2_) did not affect the main structure of Si@NH_2_ significantly, which was consistent with the FT-IR characterizations.

The microscopic morphology and geometric shape of hybrid silica (Si@NH_2_) and immobilized metalloporphyrins (Si@Porp. Co&Cu, Si@Porp. Co&Zn, and Si@Porp. Co) were explored using SEM (Scanning electron microscope) and TEM (Transmission electron microscope) techniques. From SEM images of Si@NH_2_, Si@Porp. Co&Cu, Si@Porp. Co&Zn, and Si@Porp. Co in Figure 4a–d, it could be see that the hybrid silica and immobilized metalloporphyrins prepared in this work were irregular granular aggregates with rough surfaces, and the microscopic morphology of Si@NH_2_ was not obviously affected by the immobilization of metalloporphyrins. Coupled with SEM, EDS analyses demonstrated in Figure 4 and Appendix A also illustrate that all of the essential elements in hybrid silica and immobilized metalloporphyrins were detected and distributed homogeneously on the surface of Si@NH_2_, such as carbon (C), nitrogen (N), oxygen (O), silica (Si), cobalt (Co), copper (Cu), and zinc (Zn), which also provided ample evidence for the successful immobilization of metalloporphyrins and was consistent with FT-IR characterizations and XPS analyses. The irregular granular shapes of Si@NH_2_, Si@Porp. Co&Cu, Si@Porp. Co&Zn, and Si@Porp. Co were also verified through the TEM analyses. It is illustrated in Figure 5 and Appendix A that all of the hybrid silica and immobilized metalloporphyrins were irregular granular aggregates with particle sizes of about 20–50 nm. And, characteristic elements (Si, C, N, F, Co, Cu, and Zn) were also found distributed homogeneously in Si@NH_2_, Si@Porp. Co&Cu, Si@Porp. Co&Zn, and Si@Porp. Co. The analyses of the microscopic morphology and geometric shape of hybrid silica (Si@NH_2_) and immobilized metalloporphyrins were another piece of strong evidence for the successful construction of catalytic materials in this work.

The thermal stability of immobilized metalloporphyrins prepared in this work was tested through the TGA (Thermogravimetric analysis) technique under the atmosphere of air. It is demonstrated in Figure 6 that the immobilized metalloporphyrins exhibited similar thermal stability to hybrid silica (Si@NH_2_). Before 200 °C, the weight losses of Si@Porp. Co&Cu, Si@Porp. Co&Zn, Si@Porp. Co, and Si@NH_2_ were about 9%, 7%, 8%, and 10%, respectively, which occurred mainly due to desorption of the adsorbed water on them and the dehydration of hydroxyl groups (Si-OH). The major weight loss of immobilized metalloporphyrins and Si@NH_2_ occurred above 400 °C, illustrating their satisfying thermal stability, and the weight loss above 400 °C might be mainly ascribed to the oxidative degradation of organic components under the atmosphere of air. From the TGA curves in Figure 6, it can also be calculated that the mass ratios of metalloporphyrins immobilized on Si@Porp. Co&Cu, Si@Porp. Co&Zn, and Si@Porp. Co were about 5%, 3%, and 3%, respectively. The higher loading amount of metalloporphyrins in the catalytic material Si@Porp. Co&Cu might be a reasonable source for its superior catalytic performance compared to Si@Porp. Co&Zn and Si@Porp. Co, as shown in the later part of this work. As for the different loading amount of metalloporphyrins in Si@Porp. Co&Cu, Si@Porp. Co&Zn, and Si@Porp. Co, it might be because of the effect of metal centers on the reactivity of metalloporphyrins in the immobilization reaction, because all of the reaction conditions were same in the immobilization process.

The affinities of hydrocarbons and their partial oxidation products to hybrid silica (Si@NH_2_) and immobilized metalloporphyrins were investigated through contact angle measurements employing cyclohexane and its partial oxidation products (cyclohexanol and cyclohexanone) as model compounds. As demonstrated in Figure 7, because of the presence of a large number of polar hydroxyl and amino groups on the surface of Si@NH_2_, the contact angle between non-polar cyclohexane and Si@NH_2_ reached up to 29° (Figure 7a), and when cyclohexanol and cyclohexanone with higher polarity were dripped onto the surface of Si@NH_2_, the contact angles decreased to 25° and 22° (Figure 7b,c), which was consistent with the similarity–intermiscibility theory [43,44,45,46]. When metalloporphyrins Tris(perF)P(4-CH_2_Cl)PPCo (5-(4-(Chloromethyl)phenyl)-10,15,20-tris(perfluorophenyl)porphyrin cobalt(II)) and Tris(perF)P(4-CH_2_Cl)PPCu were utilized as less polar fluorocarbons [30,31,32] and immobilized on the surface of Si@NH_2_, the contact angles of cyclohexane on Si@Porp. Co and Si@Porp. Co&Cu decreased from 29° to 14° and 11°, respectively (Figure 7a,d,g). Meanwhile, the contact angles of cyclohexanol and cyclohexanone on Si@Porp. Co and Si@Porp. Co&Cu increased from 25° to 55° and 58° (Figure 7b,e,h) and from 22° to 52° and 56° (Figure 7c,f,i), respectively. Compared with Si@NH_2_, the decreased contact angles of cyclohexane on Si@Porp. Co and Si@Porp. Co&Cu were mainly ascribed the lower polarity of their surfaces after immobilization of metalloporphyrins with C-F bonds, and their low polar surfaces led to the higher contact angles of polar cyclohexanol and cyclohexanone, correspondingly. From the contact angle measurements utilizing cyclohexane and Its partial oxidation products as model compounds, it could also be seen that the lower polar hydrocarbons could access the surfaces of the targeted catalytic materials prepared in this work readily, such as Si@Porp. Co and Si@Porp. Co&Cu, and the accessibility of partial oxidation products to their surfaces with catalytic centers could be suppressed, as compared with the corresponding substrate hydrocarbons, POX-products possessed higher polarity, which was consistent with our initial design of this work to inhibit the deep conversion of POX-products in the oxidative functionalization of C-H bonds utilizing O_2_ by reducing the accessibility of POX-products to catalytic centers.

Based on the systematic characterizations above, metalloporphyrins were immobilized on surface of hybrid silica (Si@NH_2_) successfully. All of the immobilized metalloporphyrins were irregular granular aggregates with a rough surface, and the functional elements were distributed homogeneously on their surfaces. As for the metal elements with catalytic activity, cobalt (Co) was exhibited mainly as +2 and +3 valences, and copper (Cu) and zinc (Zn) possessed only +2 valence. More importantly, when metalloporphyrins with C-F bonds were immobilized on Si@NH_2_, the obtained catalytic materials could suppress the convenient accessibility of partial oxidation products to the catalyst surface with catalytic centers and with no effect on the accessibility of substrates to the catalyst surface. The catalytic materials prepared conformed to the original intention of this work.

### 3.2. Preliminary Exploration Experiments

To inhibit the deep conversion of partial oxidation products (POX-products) in C-H bonds’ functionalization utilizing O_2_ and achieve efficient and accurate hydrocarbon oxidation, the feasibility of utilizing relay catalysis of dual metalloporphyrins with C-F bonds on the surface of hybrid silica was evaluated by firstly employing the industrially important oxidation of cyclohexane as a model reaction. It is illustrated in Table 1 that at 120 °C or below 120 °C, cyclohexane could not be oxidized by molecular oxygen (O_2_) without a catalyst, and for the conversion of cyclohexane, it was still below 0.20% after a reaction at 120 °C for 8.0 h without a catalyst (Entry 3 in Table 1). Hence, for the cyclohexane oxidation utilizing O_2_, it would be very suitable and practical to assess the performance of catalytic materials at 120 °C or lower because of the absence of uncontrolled autoxidation. As metalloporphyrins tetrakis(perfluorophenyl)porphyrin cobalt (II) (T(perF)PPCo) or 5-(4-(chloromethyl)phenyl)-10,15,20-tris(perfluorophenyl)porphyrin cobalt(II) (Tris-(perF)P(4-CH_2_Cl)PPCo) were applied as homogeneous catalysts to cyclohexane oxidation utilizing O_2_ at 120 °C, the conversion was increased to 4.25% and 3.87%, respectively, with POX-products’ selectivity (hydroperoxide, alcohol, and ketone) being about 85% (Entry 6 and Entry 8 in Table 1). Both substrate conversion and POX-products’ selectivity were at a common and reasonable level in cyclohexane oxidation with the utilization of homogeneous metalloporphyrins as catalysts [26,27,47,48]. And, when tris(perF)P(4-CH_2_Cl)PPCo was immobilized on the surface of hybrid silica (Si@NH_2_) and applied as a heterogeneous catalyst to cyclohexane oxidation utilizing O_2_ (Si@Porp. Co, Entry 10 in Table 1), the selectivity towards POX-products (hydroperoxide, alcohol, and ketone) was increased from 84.8% to 91.6%, with cyclohexane conversion dropping from 3.87% to 2.22%. Both the increased selectivity and the decreased conversion resulted from the suppressed accessibility of the substrate and POX-products to catalytic sites of heterogeneous catalysis, implying the feasibility of inhibiting the deep conversion of POX-products in C-H bonds’ functionalization utilizing O_2_ by suppressing the convenient accessibility of POX-products to catalytic sites. Although immobilized tris(perF)P(4-CH_2_Cl)PPCu on the surface of Si@NH_2_ (Si@Porp. Cu) exhibited no outstanding catalytic performance similar to homogeneous T(perF)PPCu and tris(perF)P(4-CH_2_Cl)PPCu (Entry 7, Entry 9, and Entry 11 in Table 1) in cyclohexane oxidation utilizing O_2_, when tris(perF)P(4-CH_2_Cl)PPCo and tris(perF)P(4-CH_2_Cl)PPCu were immobilized on Si@NH_2_ to form a relay catalysis system (Si@Porp. Co&Cu) on the surface of Si, the POX-products’ selectivity was further increased to 96.2% from 91.6% compared to Si@Porp. Co with a single metal center as a catalyst (Entry 10, Entry 12 in Table 1). And, cyclohexane conversion was increased to 2.41% from 2.22%, too. The simultaneously increased conversion and selectivity were consistent with the relay catalysis models reported in our previous work [9,15,27]. Therefore, in consideration of the sustained increase in the selectivity of POX-products from homogenous T(perF)PPCo and tris(perF)P(4-CH_2_Cl)PPCo as catalysts to heterogeneous Si@Porp. Co, and from Si@Porp. Co to Si@Porp. Co&Cu, and the further increased substrate conversion from Si@Porp. Co as a catalyst to Si@Porp. Co&Cu, it is an intelligent and feasible strategy to inhibit the deep conversion of POX-products in C-H bonds’ functionalization utilizing O_2_ via relay catalysis of dual metalloporphyrins on the surface of some inert and easily accessible supports and achieve efficient and accurate hydrocarbon oxidation utilizing O_2_, which is consistent with the original design intention of this work.

### 3.3. Study on Catalyst Structure

#### 3.3.1. Effect of Metal Centers

With the establishment of the feasibility of inhibiting the deep conversion of partial oxidation products in C-H bonds’ functionalization utilizing O_2_ by means of relay catalysis of dual metalloporphyrins on the surface of hybrid silica, the effect of the catalyst structure on the performance was explored systematically by employing cyclohexane oxidation as a model reaction. Firstly, the effect of metal centers on the performance of the catalytic materials developed in this work was investigated. Employing 5-(4-(chloromethyl)phenyl)-10,15,20-tris(perfluorophenyl)porphyrin (tris(perF)P(4-CH_2_Cl)PP) as a ligand, immobilized dual metalloporphyrins (Si@Porp. M_1_&M_2_) were prepared to construct a relay catalysis system on the surface of hybrid silica (Si@NH_2_) utilizing cobalt (Co), copper (Cu), zinc (Zn), and nickel (Ni) as metal centers. As demonstrated in Table 2, no outstanding function was observed from the use of Si@NH_2_ as a catalyst in cyclohexane oxidation utilizing O_2_ (Entry 2 in Table 2), and when tris(perF)P(4-CH_2_Cl)PPCo was immobilized on Si@NH_2_ (Si@Porp. Co) as a catalyst with a single metal center, a conversion of 2.22% in cyclohexane oxidation could be observed with POX-product selectivity of 91.6% (Entry 3 in Table 2). This was very similar to the use of homogenous T(perF)PPCu and tris(perF)P(4-CH_2_Cl)PPCu as catalysts (Entry 7 and Entry 9 in Table 1), as Si@Porp. Cu and Si@Porp. Zn exhibited no catalytic performance in the partial oxidation of cyclohexane, which is listed in Entry 4 and Entry 5 in Table 2. It is evident from Table 2 that relay catalysis systems containing cobalt possessed superior catalytic performance in cyclohexane oxidation utilizing O_2_, especially in the utilization of Si@Porp. Co&Cu and Si@Porp. Co&Zn as catalysts, in which the cyclohexane conversion was increased to 2.41% and 2.31% from 2.22%; meanwhile, POX-products’ selectivity was intensified to 96.2% and 95.5% from 91.6% (Entry 6 and Entry 7 in Table 2). Because of the same porphyrin ligands in Si@Porp. M and Si@Porp. M_1_&M_2_, the superior catalytic performance of Si@Porp. Co&Cu and Si@Porp. Co&Zn compared to the corresponding Si@Porp. M with a single metal center was mainly attributed to the existence of relay catalysis between Co and Cu and between Co and Zn, as reported in our previous work [9,14,15,27,28,49], in which not only the oxidation intermediates cyclohexyl hydroperoxides (C_6_-OOH) were transformed to cyclohexanol and cyclohexanone catalytically instead of disordered thermal decomposition but also their oxidation ability was utilized to transform more substrates. The different catalytic performance of Si@Porp. M and Si@Porp. M_1_&M_2_, illustrated in Table 2, also indicated that in the relay catalysis of Si@Porp. Co&Cu and Si@Porp. Co&Zn, cobalt (Co) played the role of initiating the oxidation reaction, and copper (Cu) and zinc (Zn) played the role of regulating and controlling the reaction. As for the delicate difference in the catalytic performance between Si@Porp. Co&Cu and Si@Porp. Co&Zn, it might be ascribed to the different performance of Cu and Zn in relay catalysis. As demonstrated in Figure 8, the binding energy between oxidation intermediates cyclohexyl hydroperoxides (C_6_-OOH) and T(perF)PPCu (−57.20 kJ/mol) was lower than that between C_6_-OOH and T(perF)PPZn (−70.48 kJ/mol), indicating the higher ability of the metal center Cu to utilize and transform the oxidation intermediate C_6_-OOH to some extent. The higher binding energy between C_6_-OOH and T(perF)PPZn might be adverse to the catalytic transformation of C_6_-OOH. And, the binding energies herein were obtained from a quantum chemical calculation employing the Gaussian 16 program in which the calculation models were constructed employing the software GaussView 6.0 and the binding energies were the difference between the energies of the binding complexes and the total energies of the two parts before the formation of binding complexes [50]. Thus, considering the satisfying performance of Si@Porp. Co&Cu in the relay catalysis of cyclohexane oxidation, the metal centers in this work were determined to be cobalt (Co) and copper (Cu), and the target catalytic material was determined to be Si@Porp. Co&Cu with tris(perF)P(4-CH_2_Cl)PP) as the porphyrin ligand.

#### 3.3.2. Effect of Porphyrin Ligands

With the determination of the metal centers in relay catalysis, the effect of peripheral substituents in metalloporphyrins on catalytic performance was investigated, which could influence the accessibility of the substrate and partial oxidation products to the catalytic centers. As demonstrated in Table 3, in the same preparation conditions, catalytic materials possessing abundant C-F bonds exhibited better performance in cyclohexane oxidation in the utilization of O_2_ as an oxidant (Entry 5 and Entry 9 in Table 3), which coincided with our initial design to inhibit the deep conversion of POX-products in C-H bonds’ oxidation utilizing O_2_ by accelerating the disengagement of POX-products from catalytic active centers and suppressing their accessibility, both of which originated from the low affinity and wettability between polar partial oxidation products (alcohols and ketones) with less polar fluorocarbons, as shown in Figure 7 [30,31,32]. To further confirm our speculation, 5-(4-(chloromethyl)phenyl)-10,15,20-(4-chlorophenyl)porphyrin (tris(4-Cl)P(4-CH_2_Cl)PP) and 5-(4-(chloromethyl)phenyl)-10,15,20-tris(perfluorophenyl)porphyrin (tris(perF)P(4-CH_2_Cl)PP) were selected as representative ligands to investigate the affinity and wettability of cyclohexane and its partial oxidation products on the surface of Si@Porp. Co&Cu. As shown in Figure 9, with the utilization of tris(4-Cl)P(4-CH_2_Cl)PP as a ligand, the contact angles of cyclohexane, cyclohexanol, and cyclohexanone on the surface of Si@Porp. Co&Cu were 16°, 32°, and 36°, respectively. And, when tris(perF)P(4-CH_2_Cl)PP with abundant C-F bonds was utilized as a ligand, the contact angles of cyclohexane, cyclohexanol, and cyclohexanone on the surface of Si@Porp. Co&Cu became 11°, 58°, and 56°, respectively. The increased contact angles between partial oxidation products (cyclohexanol and cyclohexanone) and Si@Porp. Co&Cu with porphyrin ligands being changed from tris(4-Cl)P(4-CH_2_Cl)PP to tris(perF)P(4-CH_2_Cl)PP indicated the lower affinity and wettability of polar partial oxidation products to Si@Porp. Co&Cu using tris(perF)P(4-CH_2_Cl)PP with low polarity as a ligand. The lower affinity and wettability of polar cyclohexanol and cyclohexanone to the surface of Si@Porp. Co&Cu with tris(perF)P(4-CH_2_Cl)PP as a ligand would enhance the disengagement of partial oxidation products from catalytic active centers and suppress their accessibility, resulting in the inhibited deep conversion of partial oxidation products in C-H bonds’ functionalization with O_2_. As for nonpolar hydrocarbons as substrates, such as cyclohexane, no obvious effect was observed from the contact angles when porphyrin ligands were changed from tris(4-Cl)P(4-CH_2_Cl)PP to tris(perF)P(4-CH_2_Cl)PP, indicating that abundant C-F bonds did not influence the free accessibility of hydrocarbons to the surface of Si@Porp. Co&Cu, and C-H bond oxidation could occur smoothly without any influence. Therefore, besides relay catalysis, as illustrated in Section 3.3.1, the use of a catalytic surface possessing the capacity for product exclusion is another effective strategy to inhibit the deep conversion of POX-products in C-H bond oxidation and achieve efficient and accurate hydrocarbon functionalization by means of oxidation utilizing O_2_, such as employing tris(perF)P(4-CH_2_Cl)PP with abundant C-F bonds to exclude polar cyclohexanol and cyclohexanone. As a result, in this work, 5-(4-(chloromethyl)phenyl)-10,15,20-tris(perfluorophenyl)porphyrin (tris(perF)P(4-CH_2_Cl)PP) was determined as the target porphyrin ligand in the relay catalysis of cobalt (Co) and copper (Cu) given its higher ability to exclude polar alcohols and ketones.

#### 3.3.3. Effect of Preparation Process

As with the determination of metal centers (Co and Cu) and their ligands (tris(perF)P(4-CH_2_Cl)PP) in relay catalysis on the surface of Si@NH_2_, the preparation process of Si@Porp. M_1_&M_2_ in this work was optimized, too. It is presented in Table 3 and Appendix A that when the ratio between moles of metalloporphyrins and the mass of Si@NH_2_ was optimized from 0.08 mmol/g to 0.20 mmol/g, optimized catalytic performance was achieved at the ratio of 0.18 mmol/g for both of Si@Porp. Co and Si@Porp. Co&Cu, which ensured sufficient catalytic active centers on the catalyst surface. Employing optimized Si@Porp. Co&Cu as a catalyst and cyclohexane oxidation as a model reaction, a conversion increase from 2.41% to 4.89% was achieved with a selectivity of 89.0% towards POX-products (Entry 5 and Entry 6 in Table 3). And, when further optimizing the molar ratio between cobalt (Co) and copper (Cu) from 1:1 to 1:2 and 2:1, the cyclohexane conversion was further enhanced to 5.03% without an effect on POX-products’ selectivity in the molar ratio of 1:2 (89.3%, Entry 7 in Table 3). The superior catalytic performance above was mainly because the higher molar ratio of Cu enhanced the performance of Si@Porp. Co&Cu in relay catalysis and ensured that the oxidation intermediates C_6_-OOH were transformed smoothly to partial oxidation products with less disordered thermal decomposition, resulting in higher substrate conversion and higher POX-product selectivity.

Therefore, based on the systematic investigation of the effect of catalyst structure on performance, an optimized catalytic material of relay catalysis was obtained as Si@Porp. Co&Cu for C-H bonds’ oxidation using O_2_, and, in the optimized catalytic material, the optimized metal centers were cobalt (Co) and copper (Cu) in the molar ratio of 1:2, and the optimized porphyrin ligand was determined as 5-(4-(chloromethyl)phenyl)-10,15,20-tris(perfluorophenyl)porphyrin (tris(perF)P(4-CH_2_Cl)PP). Additionally, in the preparation of Si@Porp. Co&Cu, the ratio between moles of metalloporphyrins and the mass of Si@NH_2_ was optimized as 0.16 mmol/g. Employing cyclohexane oxidation as a model reaction, the substrate conversion in the utilization of Si@Porp. Co&Cu as a relay catalyst could reach up to 5.03%, with a selectivity of 89.3% towards POX-products. As illustrated in Table 4, the substrate conversion and POX-products’ selectivity obtained in the relay catalysis of Si@Porp. Co&Cu not only exhibited a huge advantage over the catalysis of corresponding homogenous metalloporphyrins and their combinations (Entries 2~4, Entry 8 in Table 4), but they were also higher than corresponding immobilized metalloporphyrins with single metal and their combinations (Entries 5~8 in Table 4). And, the conversion of 5.03% with a POX-products selectivity of 89.3% was also a very appealing level for industrial cyclohexane oxidation utilizing O_2_. The satisfying catalytic performance of Si@Porp. Co&Cu in hydrocarbon oxidation utilizing O_2_ verified the rationality and feasibility of our initial design to inhibit the deep conversion of partial oxidation products in C-H bonds’ functionalization utilizing O_2_ and to achieve efficient and accurate hydrocarbon oxidation through relay catalysis of dual metalloporphyrins on the surface of hybrid silica possessing the capacity for product exclusion. It also presents a promising and applicable oxidation model for oxidative functionalization and the efficient utilization of hydrocarbons.

### 3.4. Relay Catalysis Study

As demonstrated and discussed in 3.3, the relay catalysis between cobalt (Co) and copper (Cu) was an essential aspect of C-H bonds’ oxidation in the utilization of Si@Porp. Co&Cu as a catalyst, in which oxygen-containing intermediate products alkyl hydroperoxides, such as C_6_-OOH, were transformed to partial oxidation products, such as alcohols and ketones, catalyzed by the second metal center Cu in place of disordered thermal decomposition. In addition, the inherent oxidability of alkyl hydroperoxides was utilized to oxidize more C-H bonds, resulting in the inhibited deep conversion of partial oxidation products in C-H bonds’ functionalization utilizing O_2_ and the achievement of efficient and accurate hydrocarbon oxidation. To check the occurrence of relay catalysis, cyclohexyl hydroperoxide C_6_-OOH obtained from cyclohexane oxidation without a catalyst was utilized as a representative hydroperoxide, and cyclopentane was employed as a substrate containing C-H bonds to conduct the confirmatory research. As demonstrated in Table 5, when cyclopentane was mixed with C_6_-OOH in cyclohexane and kept stirring at 60 °C for 6.0 h without a catalyst, no conversion of C_6_-OOH was observed, and no oxidation products of cyclopentane were detected. But, when homogenous metalloporphyrins or heterogeneous metalloporphyrins were applied to the reaction of cyclopentane and C_6_-OOH, an obvious conversion of hydroperoxides was observed with the formation of oxidation products of cyclopentane. The confirmatory research indeed indicated the existence of relay catalysis between Co and Cu, and all of the metal centers Co, Cu, and Zn could transform alkyl hydroperoxides catalytically to partial oxidation products and enhance the oxidability of hydroperoxides to transform more substrates. Considering the catalytic performance exhibited by the metal center Co in Table 1 and the poor catalytic performance of Cu in C-H bonds’ oxidation utilizing O_2_, the major function of the second metal Cu was to catalyze the transformation of oxygen-containing intermediate products alkyl hydroperoxides produced in C-H bond oxidation utilizing O_2_ through the catalysis of the metal center Co and to boost the C-H bonds’ oxidation in the utilization of alkyl hydroperoxides as oxidants. And, the activation of O_2_ and insertion of the element oxygen into C-H bonds were dominantly achieved through the catalysis of Co sites, which was confirmed through UV-Vis absorption spectra (Figure 10). When Tris(perF)P(4-CH_2_Cl)PPCo, Tris(perF)P(4-CH_2_Cl)PPCu, and Tris(perF)P(4-CH_2_Cl)PPZn were dissolved in cyclohexane under the atmosphere of air, the typical UV-Vis absorption peaks of Tris(perF)P(4-CH_2_Cl)PPCo at around 410 nm and 530 nm decreased gradually along with time; meanwhile, its UV absorption peak at around 432 nm exhibited an increase along with time, which might be mainly attributed to the combination of O_2_ in air with Co spontaneously. But, no change was observed from the UV-Vis absorption spectra of Tris(perF)P(4-CH_2_Cl)PPCu and Tris(perF)P(4-CH_2_Cl)PPZn in cyclohexane under the atmosphere of air over time. As for the decrease in the change of the UV-Vis absorption with time, it might be because the concentration of metalloporphyrins became lower as time went on. From the different spontaneities of O_2_ to be bound by Co, Cu, and Zn sites illustrated in the UV-Vis absorption spectra in Figure 10, it can be clearly concluded that in the relay catalysis of Co and Cu, the function of Co sites was to activate O_2_ and to insert the oxygen element into C-H bonds, and Cu sites could not activate O_2_ efficiently when their function was mainly to enhance the transformation of oxygen-containing intermediate products hydroperoxides catalytically. For the enhanced catalytic conversion of alkyl hydroperoxides, their disordered thermal decomposition was suppressed effectively, resulting in the inhibited deep conversion of partial oxidation products and the achievement of efficient and accurate hydrocarbon oxidation utilizing O_2_.

Based on the systematic investigation into the relay catalysis of C-H bonds’ oxidation utilizing O_2_ as an oxidant, employing Si@Porp. Co&Cu as a catalyst, and employing cyclohexane as a model substrate, the main path of relay catalysis in C-H bonds’ oxidation is illustrated in Figure 3. Through the first stick of the relay catalysis (red arrow), substrates with C-H bonds were transformed to the hydroperoxides R-OOH utilizing O_2_ as an oxygen source through the catalysis of the metal Co. Then, in the second stick, the oxidation intermediates R-OOH were transformed to the partial oxidation products alcohols (R-OH) and ketones (R=O) catalyzed by both the Co center and the Cu center, and in the catalytic transformation of R-OOH, fresh R-H could be oxidized by R-OOH through the catalysis of Co and Cu, as illustrated in Table 5. Clearly, the relay catalysis undertaken by the Cu site, which exhibited poor performance in C-H bonds’ oxidation utilizing O_2_ as an oxidant, intensified the catalytic transformation of oxidation intermediates R-OOH instead of the disordered thermal decomposition, which could reduce the amount of free radicals in the oxidation reaction and decrease the convenient accessibility of POX-products to active free radicals definitely, resulting in the inhibited deep oxidation and higher POX-product selectivity. In addition, copper (Cu) sites also enhanced the utilization of the oxidability of R-OOH to transform more C-H bonds, resulting in the higher conversion. Thus, besides the suppressed accessibility of polar partial oxidation products to catalytic centers induced by the abundant C-F bonds, the relay catalysis between Co and Cu was also an important source for the outstanding catalytic performance of Si@Porp. Co&Cu in the oxidative functionalization of C-H bonds with O_2_.

### 3.5. Further Optimizations

Upon determining the optimized catalyst in inhibiting the deep conversion of partial oxidation products (POX-products) efficiently, the reaction conditions of C-H bonds’ oxidation utilizing O_2_ as an oxidant in the employment of Si@Porp. Co&Cu as a catalyst were optimized further by modifying the catalyst amount, O_2_ pressure, and reaction time with the purpose of achieving efficient and accurate oxidation and utilization of hydrocarbons. And, in the optimization experiments, cyclohexane was utilized as a model substrate. As demonstrated in Table 6, Appendix A, optimized cyclohexane conversion and POX-product selectivity were obtained in the catalyst amounts of 0.16 g/mol, 0.18 g/mol, and 0.18 g/mol (catalyst/substrate), respectively, in the employment of Si@Porp. Co&Cu, Si@Porp. Co, and Si@Porp. Co&Zn as catalysts. Especially for Si@Porp. Co&Cu as a catalyst, the cyclohexane conversion climbed up to 5.27% with the POX-product selectivity of 92.3% and the catalyst amount of 0.16 g/mol (catalyst/substrate). And, no positive result was obtained with the further increase in the catalyst amount. Thus, in the C-H bonds’ oxidation utilizing O_2_ as an oxidant and catalyzed by Si@Porp. Co&Cu in this work, the catalyst amount was chosen as 0.16 g/mol (catalyst/substrate). In the O_2_ pressure optimization, as demonstrated in Table 7, Appendix A, optimized catalytic results were obtained under the pressure of 1.0 MPa when using Si@Porp. Co&Cu, Si@Porp. Co, and Si@Porp. Co&Zn as catalysts. Upon further increasing the O_2_ pressure, the oxidative degradation of catalytic materials might occur significantly, resulting in poor reaction outcomes. As for the effect of reaction time, all of the substrate conversions in cyclohexane oxidation utilizing Si@Porp. Co&Cu, Si@Porp. Co, or Si@Porp. Co&Zn as catalysts were increased by extending the reaction time, but the POX-product selectivity decreased with the extended reaction time, as illustrated in Table 8, Appendix A. Thus, to strike a balance between conversion and selectivity and to obtain a satisfying POX-product selectivity in a higher conversion, 8.0 h was chosen as the optimized reaction time in this work. Thus, an optimized catalytic system for oxidation of C-H bonds with O_2_ was developed in this work based on the systematic optimization of the reaction conditions, in which the important cyclohexane oxidation in the chemical industry was utilized as a model reaction and Si@Porp. Co&Cu was determined to be the optimized catalytic material in relay catalysis. After reacting for 8.0 h at 120 °C and 1.0 MPa O_2_, the cyclohexane conversion reached up to 5.27% with a POX-product selectivity of 92.3%. Both the conversion and POX-product selectivity were at a very competitive level in the industrial oxidation of cyclohexane with O_2_, which was also a powerful support for the correctness and rationality of our initial design in this work to inhibit the deep conversion of POX-products in C-H bonds’ functionalization utilizing O_2_ and to achieve efficient and accurate hydrocarbon oxidation via relay catalysis of dual metalloporphyrins immobilized on the surface of hybrid silica possessing a capacity for product exclusion.

### 3.6. Apparent Kinetic Study

The effect of Si@Porp. Co&Cu on the reaction activation energy in C-H bonds’ oxidation was investigated through an apparent kinetic study employing cyclohexane oxidation with O_2_. In order to facilitate the study in apparent kinetics, reactant O_2_ was supplied in excess under the constant pressure of 1.0 MPa to eliminate the effect of O_2_ concentration on the reaction rate. To eliminate the effect of mass transfer, 600 rpm was selected as the stirring rate in the apparent kinetic study. As for the catalyst amount, the optimized catalyst amount obtained in 3.5 was utilized in the apparent kinetic study, which could be regarded as a constant. Thus, the reaction rate could be treated as the function of the cyclohexane concentration only. On the basis of the above conditions, apparent zero-order kinetics, first-order kinetics, and second-order kinetics were employed to fit the relationship between substrate conversion and reaction time in cyclohexane oxidation through relational expressions kt = C_A0_ × X_A_ (zero-order), kt = ln(1/(1 − X_A_)) (first-order), and kt = X_A_/(C_A0_ × (1 − X_A_)) (second-order) [51]. The cyclohexane oxidations in the absence of a catalyst and in the presence of the catalysts Si@Porp. Co and Si@Porp. Co&Cu were adopted to conduct the apparent kinetic study. In the relational expressions above, k is the rate constant in the corresponding reaction temperature, t is the reaction time (h), C_A0_ is the initial concentration of cyclohexane (mol/L), and X_A_ is the substrate conversion at time t. As illustrated in Figure 11, Table 9, Appendix A, the cyclohexane oxidation utilizing O_2_ in this work conformed to the apparent zero-order kinetics model with the highest correlation coefficient, in which the relationship between substrate conversion and reaction time can be expressed as kt = C_A0_ × X_A_. The obtained apparent zero-order kinetics model in this work was also consistent with the essence of the solvent-free C-H bond oxidation in this work, in which the change in substrate concentration was too small to affect the reaction rate obviously. Through the apparent zero-order kinetic study of cyclohexane oxidation utilizing O_2_, rate constants at different temperatures in the absence of a catalyst and catalyzed by Si@Porp. Co and Si@Porp. Co&Cu could be calculated from the slope of the linear fitting between C_A0_·X_A_ and t, which is demonstrated in Table 9. Then, the apparent activation energy for cyclohexane oxidation without a catalyst and catalyzed by Si@Porp. Co and Si@Porp. Co&Cu were calculated through the Arrhenius equation: lnk = −(Ea/R) × (1/T) + lnk_0_ [51]. As illustrated in Table 9, the apparent activation energy in cyclohexane oxidation utilizing O_2_ in the absence of a catalyst was 162.4 kJ/mol, and when Si@Porp. Co and Si@Porp. Co&Cu were employed, the apparent activation energy was decreased to 79.1 kJ/mol and 71.8 kJ/mol, respectively. The ability of Si@Porp. Co and Si@Porp. Co&Cu to decrease the reaction activation energy in cyclohexane oxidation was consistent with their catalytic performance. The superior catalytic performance of Si@Porp. Co&Cu originated from its excellent ability to decrease the activation energy in C-H bond oxidation to some extent, especially for the higher substrate conversion. Obviously, the apparent kinetic study not only provided a deep understanding of the catalytic performance of Si@Porp. Co&Cu in C-H bond oxidation utilizing O_2_ but also validated the successful construction of a catalytic system to achieve efficient oxidation and utilization of hydrocarbons in this work.

### 3.7. Substrate Scope Study

Next, the strategy to inhibit the deep conversion of partial oxidation products in C-H bonds’ functionalization utilizing O_2_ and realize efficient and accurate C-H bond oxidation by means of relay catalysis of dual metalloporphyrins on the surface of hybrid silica was applied to the oxidation of other hydrocarbons possessing C-H bonds. As shown in Table 10, the substrates were firstly extended to other representative cycloalkanes, such as cyclopentane (C_5_H_10_), cycloheptane (C_7_H_14_), cyclooctane (C_8_H_16_), and cyclododecane (C_12_H_24_). Satisfying substrate conversion and POX-product selectivity was achieved for all of the representative cycloalkanes. In the optimized reaction conditions obtained in 3.5, the conversion of C_5_H_10_, C_7_H_14_, C_8_H_16_, and C_12_H_24_ reached up to 3.57%, 10.9%, 25.2%, and 29.4%, respectively, with selectivity of 95.9%, 93.1%, 94.8%, and 99% towards POX-products. In Table 10, it can also be observed that with the increase in the carbon atoms in cycloalkanes, both the substrate conversion and selectivity towards partial oxidation products increased gradually, implying the excellent substrate compatibility of the oxidation strategy developed in this work. When alkyl aromatics were utilized as substrates, such as toluene, ethylbenzene, isopropyl benzene, and their derivatives, the optimized catalyst Si@Porp. Co&Cu exhibited acceptable substrate compatibility, too, as illustrated in Table 11. In the oxidation of toluene and its derivatives with N-hydroxyphthalimide (NHPI) as a co-catalyst, a conversion range of 12.9% to 39.0% was obtained under solvent-free conditions. And, the oxidation products covered benzylic hydroperoxides, benzylic alcohols, benzylic aldehydes, and benzylic acids. For example, in the oxidation of toluene, a conversion of 25.6% was obtained under the solvent-free condition with a selectivity of 30.2%, 5.6%, 23.7%, and 40.5% towards the oxidation products hydroperoxide, alcohol, aldehyde, and acid (Entry 1 in Table 11). It is also demonstrated in Table 11 that in contrast to the oxidation of toluene and its derivatives, in the oxidation of ethylbenzene and its derivatives, no co-catalyst (NHPI) was needed, and the oxidation products mainly focused on benzylic ketones with a conversion range of 25.7% to 45.1%. For example, when 1-ethyl-4-nitrobenzene was used as a substrate, a conversion of 40.2% was acquired, and the selectivity towards the corresponding hydroperoxide, alcohol, ketone, and acid reached 8.8%, 8.9%, 77.6%, and 4.7%, respectively (Entry 11 in Table 11). Sequentially, when isopropyl benzene and its derivatives were oxidized by O_2_ utilizing Si@Porp. Co&Cu as the catalyst, the major products became corresponding hydroperoxides in the conversion range of 22.2% to 43.3%, which can be utilized widely as raw materials in syntheses of phenol and its derivatives and as important industrial oxidants in the conversion of olefins to prepare epoxides [52,53,54]. In 1-isopropyl-4-nitrobenzene oxidation, the selectivity towards hydroperoxide reached up to 94.6% in the conversion of 22.2% (Entry 16 in Table 11). Therefore, based on the comprehensive study of substrate compatibility above, it was very convincing that the protocol developed in this work inhibited the deep conversion of partial oxidation products in C-H bonds’ functionalization utilizing O_2_, and it possessed great potential to be applied in the oxidation and utilization of hydrocarbons with O_2_ with high efficiency and high selectivity. And, the satisfying substrate compatibility also proved the correctness and feasibility of the initial idea of this work to inhibit the deep conversion of partial oxidation products in C-H bonds’ functionalization utilizing O_2_ by means of relay catalysis of dual metalloporphyrins on the surface of hybrid silica possessing the capacity for product exclusion and to realize efficient and accurate C-H bond oxidation.

### 3.8. Mechanism Study

The reaction mechanism of C-H bonds’ oxidation utilizing O_2_ by means of the relay catalysis of Si@Porp. Co&Cu was investigated systematically in this work, too, and it is demonstrated in Figure 4 in referencing some of the relevant literature [12,22,55,56,57,58,59,60,61,62]. To confirm the speculated mechanism, the nature of the free radical process in C-H bonds’ oxidation utilizing O_2_ in this work and the major reactive species were explored in detail. Firstly, the free radical process was verified through quenching experiments. It is illustrated in Table 12 that when bromochloroform (CBrCl_3_) was utilized as the quenching reagent of carbon-centered radicals [63,64,65] and diphenylamine (Ph_2_NH) was employed as the quenching reagent of oxygen-centered radicals [64,65], the conversion during the oxidation of cyclohexane dropped sharply from 5.27% to 1.25% and 0.17%, respectively, implying that the C-H bonds’ oxidation utilizing O_2_ in this work was a free-radical process. And, in the utilization of CBrCl_3_ as a quenching reagent, both bromocyclohexane (Appendix A) and chlorocyclohexane (Appendix A) were detected in the GC-MS analysis, powerfully verifying the existence of carbon-centered radicals (cyclohexyl radical, C_6_H_11_·). Besides bromocyclohexane, *tert*-butanol was also detected in the GC-MS analysis when *tert*-butyl bromide was utilized as a quenching reagent in cyclohexane oxidation utilizing O_2_ catalyzed by Si@Porp. Co&Cu, illustrating the simultaneous occurrence of carbon-centered radicals (cyclohexyl radical, C_6_H_11_·) and oxygen-centered radicals (hydroxyl radicals, HO·) (Appendix A). To explore in greater detail the radical species in this work, electron paramagnetic resonance (EPR) measurement was also carried out utilizing 5,5-dimethyl-1-pyrroline N-Oxide (DMPO) as a radical capture reagent. It is demonstrated in Figure 12 that when cyclohexane was utilized as a model substrate possessing C-H bonds, the main radical species were carbon-centered radicals (cyclohexyl radical, C_6_H_11_·), peroxide-centered radicals (cyclohexyl peroxide radical, C_6_H_11_OO·), and oxygen-centered radicals (cyclohexyl oxygen radical, C_6_H_11_O· and hydroxyl HO·), which is in good agreement with the radical species reported in relevant documents [1,11,22,66,67]. Secondly, in the determination of the major radical species, the activation of O_2_ and the corresponding key intermediates were explored and verified, too. It is illustrated in Figure 10 that in the relay catalysis system of Si@Porp. Co&Cu, molecular O_2_ was primarily activated by cobalt (Co), not copper (Cu). Thus, the key intermediates formed by porphyrin cobalt and molecular O_2_ were investigated utilizing T(perF)PPCo (5,10,15,20-tetrakis(perfluorophenyl)porphyrin Co(II)) as a model metalloporphyrin in this work. As presented in Figure 13, key adducts between T(perF)PPCo and O_2_ (Porp. Co-OOH and [Porp. Co=O${]}_{⊕}^{▪}$) were detected in ESI-MS analyses, which is in agreement with the adduct patterns between metalloporphyrins and O_2_ reported in the relevant literature [57,62,68]. This was also strong evidence for the speculated mechanism process in Figure 4. Thirdly, it was discussed in 3.4 that in the relay catalysis between cobalt (Co) and copper (Cu) and between cobalt (Co) and cobalt (Co), both Co and Cu were employed to transform the oxidation intermediates alkyl hydroperoxides catalytically and to utilize them to transform more C-H bonds. Therefore, the interaction patterns between alkyl hydroperoxides and Co and between hydroperoxides and Cu were investigated utilizing the cyclohexyl hydroperoxides T(perF)PPCo and T(perF)PPCu as model compounds through quantum chemical calculation [50]. As illustrated in Figure 14 and Appendix A, when α-O in cyclohexyl hydroperoxides interacted with metalloporphyrins, the resultant adducts possessed lower binding energies for both T(perF)PPCo and T(perF)PPCu, illustrating that in the relay catalytic transformation of alkyl hydroperoxides by Co and Cu, the dominant configurations in the interaction between alkyl hydroperoxides and metalloporphyrins were M-α-OOH.

Thus, based on the investigation and confirmation of the free-radical nature of C-H bonds’ oxidation utilizing O_2_, the major radical species, and the key intermediates in relay catalysis, the speculated transformation path of C-H bond oxidation utilizing O_2_ catalyzed by Si@Porp. Co&Cu presented in Figure 4 was divided into four parts based on some of the relevant literature [12,22,55,56,57,58,59,60,61,62]. The first part was the classic hydroxyl rebound mechanism (I, black arrow) [69,70,71], in which the activation of O_2_ was achieved through the catalysis of Co(II) in the porphyrin cobalt unit of Si@Porp. Co&Cu and catalytically active species Co(III)-OO· and [Co(IV)=O${]}_{▪⊕}$ were formed. Both Co(III)-OO· and [Co(IV)=O${]}_{▪⊕}$ could abstract the H atom from C-H bonds, generating carbon-centered radicals (R·). By abstracting the H atom by [Co(IV)=O${]}_{▪⊕}$ from C-H bonds, the formed hydroxyl HO· in [Co(IV)-OH] could rebound to R·, generating alcohol R-OH and realizing oxygen element transfer to C-H bonds. The hydroxyl rebound path was the first source of alcohols, which could be oxidized to corresponding ketones further. Besides the hydroxyl rebound mechanism, some carbon-centered radicals (R·) could escape from the species pair formed by R· and [Co(IV)-OH] and react with O_2_ to produce alkyl peroxide radicals (ROO·), as shown in part II of the speculated reaction mechanism (Figure 2, red arrow). In the presence of the abundant C-H bonds around them, alkyl peroxide radicals (ROO·) would abstract the H atom to produce alkyl hydroperoxides (ROOH), which could transform more C-H bonds to oxygen-containing products through the catalysis of both the Co site and the Cu site. Compared with the transformation of ROO· and ROOH without a catalyst, shown in part III of Figure 4 (pink arrow), the catalytic transformation undertaken by Co and Cu in part II not only ensured that the catalytic circulate proceeded continuously but also the promotion of more C-H bonds being oxidized, resulting in the higher conversion and higher selectivity in hydrocarbon oxidation. In part III of the reaction mechanism (pink arrow), it was the intramolecular elimination of ROO· and ROOH without a catalyst, in which HO· and H_2_O were eliminated with ketone as a major product. The main objective of the relay catalysis employing Si@Porp. Co&Cu as a catalyst in this work was to transform ROO· and ROOH catalytically and to diminish the conversion of ROO· and ROOH without a catalyst, as presented in part III of Figure 4, because some free-radical chain propagation might be terminated in part III. As for part IV of the reaction mechanism in C-H bonds’ oxidation utilizing O_2_ through the catalysis of Si@Porp. Co&Cu, it was the classic Russell rearrangement mechanism (blue arrow) [72,73,74], in which corresponding alcohol, ketone, and O_2_ were produced in a molar ratio of 1:1:1 from two ROO·, and the free-radical chain propagation was terminated in this process, too.

As a brief summary of above mechanism study, the four parts in Figure 4 connected with each other closely and existed simultaneously in C-H bonds’ oxidation utilizing O_2_ through the catalysis of Si@Porp. Co&Cu. The relay catalysis undertaken by the Co site and the Cu site in part II guaranteed that the C-H bonds’ oxidation proceeded smoothly and continuously and avoided the termination of free-radical chain propagation and the disordered thermal decomposition of ROOH, which was an important guarantee besides obtaining a catalytic surface with the capacity for product exclusion to achieve efficient and accurate C-H bond oxidation. The systematic and detailed mechanism study also provided valuable guidance for our following work to construct more efficient catalytic system to inhibit the deep conversion of POX-products in C-H bonds’ functionalization utilizing O_2_ and to achieve efficient and accurate C-H bond oxidation.

### 3.9. Literature Comparison

At last, the strategy to inhibit the deep conversion of partial oxidation products in C-H bonds’ functionalization utilizing O_2_ and achieve efficient and accurate hydrocarbon oxidation through relay catalysis of Si@Porp. Co&Cu on the surface of hybrid silica possessing the capacity for product exclusion was compared with some current reports (2021–2023). And, the industrially significant partial oxidation of cyclohexane was utilized as a model reaction to investigate the advantage of oxidation methods in this work and in the literature from the aspects of reaction conditions, substrate conversion, POX-product selectivity, accuracy and rationality of quantitative methods, and the reasonability of experimental data. As demonstrated in Table 13, because of the difference in reaction conditions and quantitative methods, the obtained cyclohexane conversion and POX-product selectivity (cyclohexanol, cyclohexanone, and hydroperoxide) exhibited a greater diversity. The cyclohexane conversion ranged from 5% to 40% with a selectivity range of 60% to 99% towards partial oxidation products. The relatively wide range in the reported experimental data could mainly be attributed to the different reaction conditions and quantitative methods, especially the reaction temperature. As shown in Table 1, at 130 °C, the conversion in cyclohexane oxidation without a catalyst could reach up to 0.44%, implying the occurrence of obvious autoxidation. It was very pleasing in the literature comparison that the quantitative method employed in C-H bonds’ oxidation is becoming more and more comprehensive, accurate, and reasonable. From the utilization of GC (Gas chromatography analysis) only to GC + chemical titration and to GC + HPLC, both the oxidation products with low boiling-points (liquid products) and the products with high boiling points (solid products) were analyzed quantitatively, providing a series of reasonable experimental data. Current studies on cyclohexane oxidation not only have made a significant contribution to achieving efficient and accurate cyclohexane oxidation but have also promoted deeper understanding into mechanisms of C-H bond oxidation tremendously, which has become a solid foundation to achieve C-H bond oxidation more and more efficiently and selectively.

Based on the discussion and analyses above, the major advantages of the strategy developed for C-H bonds oxidation utilizing O_2_ in this work employing Si@Porp. Co&Cu as catalyst could be summarized as follow. (1) Novel and effective oxidation model: Relay catalysis on surface with the capacity for product exclusion. (2) Mild reaction conditions: 120 °C, solvent-free and co-catalyst-free. (3) High selectivity towards POX-products with acceptable conversion: Selectivity of 92% in conversion of 5.27% for cyclohexane. (4) Comprehensive quantitative analysis on oxidation products: GC + HPLC. (5) Reasonable experimental data, in which the effect of substrate evaporation on the conversion was eliminated effectively, especially in cyclohexane oxidation, and the amount of deep oxidation products (Carboxylic acids) and their effect on the POX-products selectivity were analyzed and calculated exactly. Thus, this work not only could serve as a novel and practical strategy to inhibit the deep conversion of POX-products in C-H bonds oxidation, and achieve efficient and accurate hydrocarbons oxidation through the relay catalysis of Si@Porp. Co&Cu on surface possessing the capacity for product exclusion, but also afforded a systematic and reasonable research method in C-H oxidation for both of theoretical research and industrial application.

## 4. Conclusions

In summary, fluorine-containing metalloporphyrins Tris(perF)P(4-CH_2_Cl)PPCo and Tris(perF)P(4-CH_2_Cl)PPCu were immobilized on the surface of hybrid silica (Si@NH_2_) successfully to construct a relay catalysis system on the surface with product exclusion ability (Si@Porp. Co&Cu). All of the representative catalytic materials were characterized and confirmed thoroughly in this work by the means of various characterization techniques. Employing C-H bonds’ oxidations as target reactions, the deep conversion of partial oxidation products (POX-products) could be inhibited efficiently, and the substrate conversion and POX-products’ selectivity were improved simultaneously through relay catalysis of Si@Porp. Co&Cu. For the important cyclohexane oxidation with O_2_, the conversion was improved from 3.87% to 5.27%, with the selectivity increasing from 84.8% to 92.3%. The outstanding performance of Si@Porp. Co&Cu as a catalyst in C-H bonds’ oxidation mainly originated from the suppressed accessibility of polar POX-products (alcohol and ketone) to catalytic sites on the lower polar and fluorine-containing surface and the relay catalysis between Co and Cu to utilize the oxidation intermediates alkyl hydroperoxides to transform more C-H bonds. The product exclusion ability, relay catalysis process, apparent kinetic study, substrate scope, and reaction mechanism were also investigated in depth in this work. Compared with the catalytic systems reported in the current literature, the major advantages of the strategy for C-H bonds’ oxidation utilizing O_2_ developed in this work were (1) a novel and effective oxidation model, (2) mild reaction conditions, (3) high POX-product selectivity with acceptable conversion, (4) comprehensive quantitative analysis of oxidation products, and (5) reasonable experimental data. This work not only presented a novel and practical strategy for C-H bond functionalization utilizing O_2_ to inhibit the deep conversion of POX-products and achieve efficient and selective hydrocarbon oxidation; it also provided a valuable protocol for other important chemical transformations to avoid over-reaction and achieve high selectivity.

## Data Availability

Data will be made available upon request.

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
