# Peer review of "Efficient Inhibition of Deep Conversion of Partial Oxidation Products in C-H Bonds’ Functionalization Utilizing O2 via Relay Catalysis of Dual Metalloporphyrins on Surface of Hybrid Silica Possessing Capacity for Product Exclusion"

_biomimetics, 2024, doi:10.3390/biomimetics9050272_

Round 1

Reviewer 1 Report

Comments and Suggestions for Authors

Review Report:

In their manuscript Zhang et al. demonstrated an efficient strategy to inhibit deep conversion of partial oxidation products in C-H functionalization using O2 via relay catalysis of dual metalloporphyrin on surface of hybrid silica. Various analytical methods, such as FTIR, XRD, XPS were employed to characterize relay catalysis. Through their work the authors were able to improve the cyclohexane oxidation from 3.8% to 5.3% together with an improvement of selectivity by ~7%.

This study is important in the context of efficient conversion of hydrocarbons by minimizing the deep conversion of partial oxidation products. The manuscript is thorough and fits with the journal scope. I recommend publication of the manuscript after a minor modification. My comments are copied below. I use the following abbreviation, P-page number.

Specific Comments:

1. P7: While explaining the FTIR vibrational transitions of metalloporphyrin immobilized on hybrid silica, please compare with the transitions resulting from isolated metalloporphyrin to comprehend the changes resulting from the surface interactions.

2. P9-Figure2: I encourage the authors to provide a background subtracted XPS spectra. In addition, I see two different backgrounds between the 1st and 2nd segments. What’s the source of this difference?

3. P23-Figure10a: In these spectra, do the authors see any kinetics associated with the band around 220 nm? What’s the source of this transition?

Reviewer 2 Report

Comments and Suggestions for Authors

This paper is a comprehensive analysis of a novel method for C-H bond functionalization using O2 to prevent the formation of POX-products. The level of detail and the number of experiments carried out shows that a lot of effort has been made here and I commend the authors for this. The authors have found catalytic materials that seems to work and have briefly compared them to other catalytic systems in the literature. At the same time, I am left wondering what the true impact of the findings of this study is.

I do find this paper too long and detailed, which makes it hard to read, and even more difficult to evaluate all data and conclusions with reasonable care. The authors could improve by summarizing the data and highlighting the main findings. The most important results are lost in the details, which does not help the authors. I worry that this could reduce the impact of this paper. Overall, I would stress that the authors should focus only on the key aspects which makes this approach to inhibiting deep conversion of partial oxidation products so important, and substantially reduce the details presented herein. I think the manuscript is not ready for publication yet, and some of the other concerns I have are as follows:

1. The introduction section has too many details. It looks more like a review but should be concise. It is hard to follow the main argument of this paper when there are so many details of previous literature results. The authors should think about rewriting the introduction to include only the essential points from previous work, and to make sure that whatever is written clearly supports the motivation for this paper.

2. In scheme 1, the transformation route of CH bond oxidation is presented. In the text, several papers (6 in total) including some of the author’s own detailing a very similar study to this one are presented. Could the authors show in the scheme the references relevant to each step in the transformation?

3. The authors state that “representative elements of metalloporphyrins (N, Co, Cu, Zn and F) were observed obviously” based on the XPS data. They have marked these areas in Figure 2. In figure 2a, the areas for Zn 2p, Cu 2p and Co 2p seem to be noise. I suppose these areas are enlarged in the following panels but this is not clearly stated (please correct this). In these areas, there is a lot of variation in the data points. For example, in panel d the authors show Cu2+ and satellite peaks but I cannot see how these peaks can be derived from the data. Except maybe for panel f, I am not persuaded by the fittings. It would be helpful if the authors explain how they did their fits of the data and how they can support the claim that these elements have been seen “obviously” at least by XPS.

4. Regarding the PXRD data, what are the extra peaks around 18, 35 and 45 deg?

5. It's possible I may have overlooked this, but the authors do not appear to refer to the Tris(perF)P(4-CH2Cl)PPCo when they talk about figures 1&3.

6. The authors say that the reason why Si@Porp, Co&Cu and others have different loading amounts is because the metal centers affect how reactive the metalloporphyrins are in the immobilization reaction. Please explain this statement more. Also, before this comment the authors say that higher loading on Si@Porp Co&Cu could be related to its better catalytic performance. I don't understand what performance the authors mean and how much difference a 5% loading vs 3% loading would make for catalytic performance? Are they actually referring to results appearing later in the manuscript?

7. The authors showed some contact angle measurements. Some of these angles are almost the same (e.g. 29 deg vs 25 deg vs 22 deg). It looks like cyclohexanol and cyclohexanone have no real difference, and maybe this is one of the main ideas. The difference between cyclohexane and the other two is obvious. Could the authors say something about how reliable these measurements are?

8. The authors have not provided any uncertainties for the catalysis studies (e.g. conversion). These uncertainties are essential and should be included in the measurements – along with that, I do not see any information about the number of times the reactions were repeated.

9. The authors used quantum chemical methods to get theoretical binding energies. (Fig 8). The authors, however, did not provide any information about how they built these models and how they computed the energies. This information needs to be included.

10. The authors state in table 3 that the inclusion of more C-F bonds improved oxidation. However, I think that most of the results are not very different unless the Co/Cu ratio is varied. Did the authors attempt this change with all the other entries?

11. The authors should clarify the following statement: “the conversion of 5.03% with a POX-products selectivity of 89.3% was also a very appealing level for industrial cyclohexane oxidation utilizing O2” … in comparison to what? I might have overlooked it, but I’m not sure what the current benchmark is to evaluate whether this is good. Also, entries 5 and 7 seem to be interesting too … could the authors explain why entry 8 has an advantage over the other two – or is it just based on the numbers.

12. The main text does not explain Figure 10 well. For instance, the panels are not marked clearly. I am not sure what I am seeing. Also, the peak at 530 nm stays the same - while there is a slight change for the peaks at 410 and 435 nm (only in the first 5-10 minutes). What causes this difference in how these peaks change over time? Or is it not a difference? Maybe the relative change is equal. Can the authors elaborate on this? Also what is the peak at 220 nm? Is it present in the Cu/Zn samples too (the x-axis is not shown at this wavelength in those ones).

13. I'm not sure how the authors decided on the reaction temperatures they used for the kinetic study (and is it normal to only use three temperature points?). Why are the temperatures different between a and c/e? Also, I'm curious if the authors would consider moving either the table or the figure related to this study to the SI because they are essentially showing the same data.

14. I think tables S11 and S12 are not referenced in the main text. Figure S6 to S9 need more descriptive figure captions explaining each part of the figure. The chemical structures shown in these figures are also poorly drawn. All figure captions must be clear and concise.

15. The SI contains many figures that are not mentioned in the main text or the SI itself. Figures that are not relevant, explained or at the very least referred to should not be included. This includes figures corresponding to characterization data (e.g. NMR spectra).

16. The authors list 5 reasons why their work is persuasive. (1) Novel: This paper seems to be a similar example of some of their previous work (e.g. Molecular Catalysis, 557, 113957, 2024 – published on March 15th of this year) which seems to include a similar approach and result with the same or similar compounds except for the hybrid silica. I do not believe the authors make reference to this paper. Is the current manuscript meant to be complimentary? If so, maybe it is the product exclusion & POX selectivity that makes it novel, but then the authors should consider reducing the details in this manuscript and focus only on these aspects. Otherwise, this study seems to be a small incremental improvement carried out roughly the same time as prior one(s). Finally, could the authors explain why points 4 and 5 are part of this list? The data analysis and experimental data are expected in the paper.

Minor issues:

The paper is very long and makes use of a lot of abbreviations. Sometimes this is unavoidable, however I feel that the authors could mitigate some confusion and make the text easier to read by using an numbering scheme for the main compounds. I find it very confusing to read full chemical names (e.g. tris(perF)P(4-CH2Cl)PPCo) or Si@Porp. Co) when the authors could use a scheme such as 1-Co 1-Cu etc for the former and some similar scheme for the latter.

Some figure captions are unclear. In figure 1, it is not clear what the overlaid boxes and dashed lines correspond to without digging through the main text. In figure 2, the panels are not clearly labelled in the figure caption. Same with Figure 4. Please add all relevant information to all figure captions.

In section 2.1, line 4, there is a spelling mistake… “as follows

In section 2.1 there is a strange gap between “…porphorin” and “copper(II)”.

In figure 1, the text overlaying the figures does not seem to properly line up with the actual figure panels. Also white text over panel e is illegible.

On page 12, please correct “in Figure 6, it was also could be calculated” … there seems to be words missing.

On page 17, there is a missing space between a degree symbol and the next word.

In table 4, compound numbers would benefit the authors to avoid formatting issues and poor visual appearance.

Table 11 is very difficult to read. Please find a way to improve the separation between the rows or reduce the size/bulkiness of the chemical structures. Perhaps this table should be moved to the SI.

Figure 13 – the overlaid structures do not seem to line up with anything and are covering the y-axis.

Table S9 in the main text should be bold.

Comments on the Quality of English Language

No comments.
